# Peripheral Nerve Denervation in Streptozotocin-Induced Diabetic Rats Is Reduced by Cilostazol

**DOI:** 10.3390/medicina59030553

**Published:** 2023-03-11

**Authors:** Kuang-Yi Tseng, Hung-Chen Wang, Yi-Hsuan Wang, Miao-Pei Su, Kai-Feng Cheng, Kuang-I Cheng, Lin-Li Chang

**Affiliations:** 1Graduate Institute of Clinical Medicine, College of Medicine, Kaohsiung Medical University, Kaohsiung 807378, Taiwan; 2Department of Anesthesiology, Kaohsiung Medical University Hospital, Kaohsiung Medical University, Kaohsiung 807378, Taiwan; 3Department of Neurosurgery, Chang Gung Memorial Hospital-Kaohsiung Medical Center, Chang Gung University College of Medicine, Kaohsiung 833253, Taiwan; 4Department of Medical Research, Kaohsiung Medical University Hospital, Kaohsiung 807378, Taiwan; 5Department of Microbiology and Immunology, College of Medicine, Kaohsiung Medical University, Kaohsiung 807378, Taiwan

**Keywords:** diabetes, cilostazol, streptozotocin, epidermal nerve fiber

## Abstract

*Background and Objective:* Our previous study demonstrated that consistent treatment of oral cilostazol was effective in reducing levels of painful peripheral neuropathy in streptozotocin-induced type I diabetic rats. As diabetic neuropathy is characterized by hyperglycemia-induced nerve damage in the periphery, this study aims to examine the neuropathology as well as the effects of cilostazol treatments on the integrity of peripheral small nerve fibers in type I diabetic rats. *Materials and Methods:* A total of ninety adult male Sprague-Dawley rats were divided into the following groups: (1) naïve (control) group; (2) diabetic rats (DM) group for 8 weeks; DM rats receiving either (3) 10 mg/kg oral cilostazol (Cilo10), (4) 30 mg/kg oral cilostazol (Cilo30), or (5) 100 mg/kg oral cilostazol (Cilo100) for 6 weeks. Pain tolerance thresholds of hind paws toward thermal and mechanical stimuli were assessed. Expressions of PGP9.5, P2X3, CGRP, and TRPV-1 targeting afferent nerve fibers in hind paw skin and glial cells in the spinal dorsal horn were examined via immunohistochemistry and immunofluorescence. *Results:* Oral cilostazol ameliorated the symptoms of mechanical allodynia but not thermal analgesia in DM rats. Significant reductions in PGP9.5-, P2X3-, CGRP, and TRPV-1-labeled penetrating nerve fibers in the epidermal layer indicated denervation of sensory nerves in the hind paw epidermis of DM rats. Denervation significantly improved in groups that received Cilo30 and Cilo100 in a dose-dependent manner. Cilostazol administration also suppressed microglial hyperactivation and increased astrocyte expressions in spinal dorsal horns. *Conclusions:* Oral cilostazol ameliorated hyperglycemia-induced peripheral small nerve fiber damage in the periphery of diabetic rats and effectively mitigated diabetic neuropathic pain via a central sensitization mechanism. Our findings present cilostazol not only as an effective option for managing symptoms of neuropathy but also for deterring the development of diabetic neuropathy in the early phase of type I diabetes.

## 1. Introduction

Diabetic peripheral neuropathy (DPN) is one of the most common chronic vascular complications in patients with diabetes mellitus (DM), with approximately 60% prevalence in either type I or type II DM in both sexes globally [1,2,3]. Long-term persistent hyperglycemia leads to deterioration of the blood–brain barrier, break down of the blood–nerve barrier in the peripheral nerve, as well as damages to Schwann cells and primary afferent nociceptors. Progressive diabetic neuropathy was initially observed in distal sensory, autonomic, and motor axons. These disruptions further lead to hyperexcitability in peripheral and central neurons, resulting in either hyper- or hyposensitivity of peripheral sensory receptors [4,5,6]. The unique neurodegenerative disorder of the peripheral sensory axons is a type of small fiber neuropathy (SFN) caused by diabetes to impair thinly myelinated Aδ-fibers, unmyelinated C-fibers, and limb epidermal nerve fibers. Diabetic length-dependent neuropathy is characterized by damage to the longest sensory axons, with the loss of distal limb epidermal axons prior to its proximal limbs. However, diabetes-induced SFN is difficult to diagnose using sensory or motor nerve examinations at early onset and is only treated according to the initial symptoms. Inadequate blood flow induces disproportionally degenerated small-fiber nerves, including nociceptors, which can ultimately lead to the degeneration of superficial spinal dorsal horn neurons due to hyperglycemia [7] or maladaptive aberrant input from the periphery [4]. SFN is commonly observed in the early stages of diabetes and prompts clinical management to prevent its progression to even more devastating complications, such as foot ulceration or cardiac dysautonomic death, with much more severe consequences.

Cilostazol is a potent phosphodiesterase inhibitor commonly known for its antiplatelet and vasodilatory properties, as well as its ability to reduce serum lipid levels [8,9]. It reportedly has no detectable effects on blood glucose levels but has been shown to be beneficial for modulating motor deterioration in patients with diabetes to a certain extent [10,11,12]. In other DM experimental models, cilostazol demonstrated the ability to reduce inflammatory burden and oxidative stress as well as protect cardiac vessels against hyperglycemia-induced injuries [10,11,13,14,15,16]. In vitro and in vivo studies have also shown the beneficial effects of the medication in reducing endothelial dysfunction and improving peripheral blood flow [10,17,18,19]. More importantly, recent reports have demonstrated the ability of oral cilostazol in ameliorating hyperglycemia-induced peripheral neuropathic pain in streptozotocin (STZ)-induced DM rats. Its apparent beneficial effects on blood flow and overall vascular health, as well as the amelioration of DPN phenotype in DM rats, suggest certain levels of neuroprotection of peripheral organs in DM subjects. This study examined changes in the expression of some afferent nociceptive nerve markers in the hind paw epidermis of STZ-induced DM rats with or without cilostazol treatments to assess the exact effects of various doses of oral cilostazol on the integrity of cutaneous nerves and functional maintenance. The effects of the medication on nociceptive nerve distributions in the paw epidermis of rats with DM were further examined in correlation with the ability of cilostazol to effectively reduce the DPN phenotype and microglial activation in the spinal dorsal horns (SDH).

## 2. Materials and Methods

### 2.1. Preparations and Administrations of Cilostazol

Cilostazol (C0737, Sigma Aldrich) was prepared in normal saline and administered daily via oral gavage at 10, 30, or 100 mg/kg starting on the third week of successful DM induction for six weeks until sacrifice (refer to Figure 1 for treatment timeline). 

### 2.2. Animals and Diabetes Induction

This study used a total of ninety adult male Sprague-Dawley rats weighing 250–300 g. All the animals were housed in plastic cages with routinely refreshed clean, soft bedding and maintained under a 12-h light-dark cycle (light cycle 7 a.m.–7 p.m. and vice versa), with access to food and water ad libitum. The use of the animals and all relevant experimental procedures described in this study were approved by the Kaohsiung Institutional Animal Care and Use Committee (Approval No.106008). Rats were divided into five groups as follows: (1) naïve (control) group, with rats that underwent surgery to expose the right femoral vein for intravenous injection of normal saline; (2) diabetes mellitus (DM) group, with rats that received a single-dose injection of 60 mg/kg streptozotocin (STZ, Sigma, St. Louis, MO, USA) via the right femoral vein; (3) DM plus 10 mg/kg cilostazol (Cilo10) group, with DM rats receiving a daily dose of oral cilostazol at 10 mg/kg via oral gavage (o.p.) for six weeks; (4) DM plus 30 mg/kg cilostazol (Cilo30) group, with DM rats receiving a daily dose of 30 mg/kg o.p. cilostazol for six weeks; (5) DM plus 100 mg/kg cilostazol (Cilo100) group, with DM rats being administered a daily dose of 100 mg/kg o.p. cilostazol for six weeks. All cilostazol treatments began on the third week, post-operation day (POD) 15 of successful hyperglycemia induction and continued for 6 weeks until sacrifice (illustrated timeline in Figure 1). Successful inductions of type I diabetes were indicated by increases in levels of random blood glucose to over 500 mg/dL within three days of operation. Blood glucose levels were assessed via the tail vein using an Accu-Chek^®^ Performa blood glucose assay kit. This femoral vein STZ injection model has a 100% success rate in producing type I DM rats, as indicated by the rapid development of hyperglycemia within 3 days of operation, with a less than 5% mortality rate so far [20]. The choices of cilostazol dosages for this experiment were based on previous publications by Naka et al. (1995) for the lower 10 and 30 mg/kg dosages [18], and Rosales et al. (2011) for the high dose [12].

### 2.3. Responsive Thresholds to Thermal and Mechanical Stimuli

Rats from all groups were subjected to plantar heat tests and electrical von Frey to test for the animals’ sensitivity toward thermal and mechanical stimuli. Rats were allowed to acclimate to their respective environments for up to 30 min prior to each session. Hypersensitivities to thermal or mechanical stimuli in the hind paw were assessed as described in our previous work and briefly as follows [20,21,22]. 

Hypersensitivities to heat were measured as the latency of hind paw withdrawal from a heat stimulus. Hind paws were set on a glass plate heated to 193 mW/cm^2^ by a directed infrared light beam through a 2 cm × 5 cm pinhole emitted from a moveable lightbox (UgoBasile Model 7370, Via Giuseppe di Vittorio, 21036 Gemonio VA, Italy) one at a time. The thermal stimulus was terminated and the heating duration recorded either by paw withdrawal from the glass plate or automatically at the 20-s cut-off time. The average of four to six tests was used to calculate the withdrawal threshold for each paw.

For assessing pain tolerance against a mechanical stimulus, the electrical von Frey facility comprised a metal mesh floor covered with a transparent plastic dome (8 cm × 8 cm × 18 cm) and a dynamic plantar esthesiometer (UgoBasile, Via Giuseppe di Vittorio, 21036 Gemonio VA, Italy). The plantar esthesiometer was set to an incremental increase of 2.5 g/sec and a maximum cut-off threshold of 50 g to measure hind paw withdrawal thresholds against a mechanical stimulus. The average withdrawal threshold for each paw was calculated by conducting three to five non-consecutive tests, with a minimum rest period of five minutes between each test for each leg.

### 2.4. Histological Samples Preparations, Immunohistochemistry, and Immunofluorescence

Selected L5 spinal cord, hind paw epidermal samples, and back trunk skin were fixed in ice-cold 4% (*w*/*v*) paraformaldehyde. Fixed spinal cord and skin samples were progressively desaturated using 10–30% (*w*/*v*) sucrose in 0.02 mol/L phosphate buffer (pH 7.4) prior to being embedded in Tissue-Tek optimal cutting temperature (O.C.T.; FSC22 Clear, Surgipath, Leica, Macquarie Park, Australia) compound in preparation for cryo-sectioning. 

For immunohistochemistry (IHC) assays, hind paw epidermal and back trunk skin samples were sectioned at 50 µm into phosphate-buffered saline with tween-20 (PBS-T, 0.05% tween) solution. The skin sections were permeabilized initially using a 50% ethanol solution for 30 min, after which they were incubated for 10 min with 0.3% hydrogen peroxide. Nonspecific bindings of secondary antibodies were blocked by an hour incubation with 2% goat serum solution dilute in PBS-T before 48-h incubation with either PGP9.5 (Millipore Sigma, AB1761, Taufkirchen, Germany), P2X3 (Neuromic, GP10108, Manhattan, NY, USA), calcitonin gene-related peptides (CGRP; Sigma Aldrich, c8189, St. Louis, MO, USA), or transient receptor potential vanilloid-1 (TRPV-1; Neuromics, RA10110, Manhattan, NY, USA) primary antibody at 4 °C. After incubating with the primary antibody, the sections were rinsed with PBS-T and then treated with biotin solution (Invitrogen, Carlsbad, CA, USA) for one hour, followed by an additional hour of incubation with avidin–biotin–peroxidase polymer (Invitrogen, Carlsbad, CA, USA). The peroxidized samples were developed via 3–5 min treatment with DAB solution (Invitrogen, Carlsbad, CA, USA), followed by counterstaining for cell nuclei with hematoxylin. Subsequently, sections were washed with PBS-T, mounted on gelatin-coated slides, and covered with Aquapolymount (Polysciences, Warrington, PA, USA). The samples were washed with PBS-T for approximately 30 min between each incubation step. Quantification and calculation of the relative number of specific target signals in skin samples were performed by manually counting trans-epidermal target-positive nerve fibers within 5 mm of the skin sections that were detectable under a high-power confocal microscope (Olympus, FV1000, Tokyo, Japan).

The spinal cord samples were sectioned at 30 µm for immunofluorescence assays and were briefly washed in PBS to remove excessive O.C.T. residues before antigen retrieval at 90 °C in citric buffered solutions (1.92 g/L, pH 6, 0.05% Tween-20) for 10 min. Nonspecific bindings of secondary antibodies were blocked by a one-hour incubation with primary antibody diluent (Tris, green; ScyTek Inc., West Logan, UT, USA) before overnight incubation with polyclonal goat anti-IbaI (Abcam, ab5076, Cambridge, MA, USA) or polyclonal goat anti-GFAP (Abcam, ab53554, Cambridge, MA, USA) primary antibody at 4 °C. Subsequently, sections were incubated with Alexa488 donkey anti-goat IgG secondary antibody (Abcam, ab150129, Cambridge, MA, USA) for two hours. Finally, the stained sections were washed with PBS and covered with FluoreGuard mounting medium (ScyTek Inc., West Logan, UT, USA). Stained sections were examined and photographed using an Olympus FluoView1000 confocal laser scanning microscope (Olympus, Tokyo, Japan). Quantification of immunofluorescence staining of the spinal cord samples was performed on ImageJ according to previous reports [20,21,22]. 

### 2.5. Statistical Analysis

Statistical analyses for data of behavioral responses and IHC results were performed using either Mann–Whitney U-test (SPSS 20.0; SPSS Inc., Chicago, IL, USA) or single-factor analysis of variance (ANOVA) test via Microsoft Excel version 16.5. Statistical significance was set at * *p* < 0.05, ** *p* < 0.01, and ☆ *p* < 0.001 for all assays. §§ indicates a significant difference at *p* < 0.001 as compared to the corresponding DM value in behavior response towards mechanical and thermal stimuli.

## 3. Results

### 3.1. Cilostazol Ameliorated Mechanical Allodynia in Diabetic Rats

Successfully induced DM rats were randomly selected for daily treatment of either 10, 30, or 100 mg/kg cilostazol via oral gavage from the third week of hyperglycemia induction for 42 days (Figure 1 timeline). Cilostazol administration did not affect the blood glucose levels in STZ-induced DM rats throughout the experimental period (Figure 1A). Moreover, diabetic rats induced by STZ exhibited reduced responsiveness to thermal stimuli and experienced a slight increase, rather than a decrease, in thermal nociceptive thresholds (Figure 1B). 

Cilostazol administration mitigated the relative blunt responses of thermal hyperalgesia and resulted in consistent thermal tolerance throughout the 8 weeks of the study in the control group (Figure 1B). In contrast, DM rats showed significantly lower levels of resistance to mechanical stimuli in the first week of hyperglycemia induction, indicating the development of mechanical allodynia in the hind limbs (Figure 1C). Significant improvements in mechanical allodynia were observed in all three cilostazol-treated groups within the first seven days of treatment (Figure 1C).

### 3.2. Cilostazol Reduced PGP9.5 Epidermal Sensory Nerve Fiber Disruptions in the Hind Paws of Diabetic Rats

PGP-9.5, also known as neuronal cytoplasmic protein 9.5, is a sensitive and common marker of nerve sheaths [23,24]. Significantly lower numbers of PGP9.5-immunolabeled nerve endings were observed in the epidermal layer of the hind paws of diabetic rats than those of the control group (Figure 2A,B). In the low-dose (10 mg/kg) cilostazol treatment group, the number of PGP9.5-positive penetrating nerve fibers did not persevere (Figure 2C,F). However, the number of PGP9.5-positive nerve fibers in the epidermal layers was reversed in DM rats administered with 30 or 100 mg/kg cilostazol (Figure 2D–F). In contrast, there were no significant differences in the number of PGP9.5-positive penetrating nerves in the epidermal layers of back trunk skin between control and diabetic rats (Figure 2G). In addition to the reduced number of PGP9.5-positive epidermal sensory nerve fibers, more swollen or shorter PGP9.5-positive nerve fibers were also observed in the epidermal layers of the hind paws of the foot pads of DM rats compared to control rats (Figure 2I,H).

### 3.3. Cilostazol Reduced P2X3, CGRP, and TRPV-1 Epidermal Nociceptive Nerve Disruptions in the Hind Paws of DM Rats

P2X3 is an ATP-dependent ion channel receptor expressed in primary afferent neurons and is commonly associated with the development of neuropathic pain [25,26]. DM rat samples showed significantly lower numbers of P2X3-immunolabeled sensory nerve endings in the intraepidermal layers of the hind paws than those of the control group (Figure 3A,B,F). There was a significant increase in the number of epidermal P2X3-positive penetrating nerve endings in the DM group receiving either 30 or 100 mg/kg cilostazol compared to those in the no-treatment DM group (Figure 3B–F). In contrast, no obvious changes in P2X3-positive nerve endings were detected in the back trunk epidermis between the groups (Figure 3G).

CGRPs are primarily localized to the C and A𝛿 sensory fibers that are known to be associated with pain processes [23,27]. The densities of CGRP-immunolabeled intraepidermal nerve endings were significantly lower in the hind paws of DM rats compared to the control group (Figure 4A,B,F). In contrast, the number of CGRP-positive nerve endings detected in the cilostazol-treated DM groups appeared comparable to those observed in the control group, but with relatively swollen and shortened penetrating nerves in the epidermal layer (Figure 4B–F). No significant differences in the number of CGRP-positive nerve endings in the dorsal trunk epidermis were detected among DM rats treated with or without cilostazol (Figure 4G).

TRPV-1, a nonselective cation channel, is normally expressed on the peripheral and central terminals of small-diameter sensory neurons and is frequently studied in conjunction with CGRP with regard to pain [28,29,30]. In the control rats, TRPV-1 expressions were largely observed in the dermis, close to the epidermal layer (Figure 5A). The hind paws of DM rats showed reduced numbers of detectable TRPV-1-positive nerve fibers (Figure 5B). Among the cilostazol-treated DM rats, the 30 and 100 mg/kg treatment groups showed relatively preserved numbers of TRPV-1 positive nerve fibers in the paw dermis (Figure 5C–F). 

### 3.4. Cilostazol Reduced Aberrant Expression of Glia Cells in the Spinal Dorsal Horn of Diabetic Rats

Compared to the control group, DM rats showed persistent hyperglycemia-induced microglial overactivation (Figure 6A,B) but reduced astrocyte expression (Figure 6D,E) in the L5 SDH. However, 100 mg/kg cilostazol administration ameliorated microglial overactivation (Figure 6C,G) and restored the reduced astrocytes expressions (Figure 6F,H) toward normal levels of SDH expression in control rats.

## 4. Discussion

The present study showed that persistent hyperglycemia in rats resulted in distal peripheral nerve degeneration in the hind paw with the total number of free nerve endings in the epidermal layer penetrating from the dermis, or free nerve endings of either purinergic or non-purinergic types decreased in DM rats. Denervation was characterized by obvious loss and truncation of penetrating afferent nerve fibers in the dermal–epidermal layers of the hind paw of diabetic rats rather than in the back trunk skin. In comparison, the apparent denervation was significantly ameliorated in hind paw samples of Cilo30 and Cilo100 treatment DM rats, as the higher dosage groups showed greater improvements in peripheral nerve distribution in diabetic rat hind paw skin. However, the loss of these free nerve endings does not completely reflect the observations representing mechanical allodynia. In this study, untreated diabetic rats showed a loss of free nerve endings in the epidermal layer while demonstrating obvious mechanical allodynia. In contrast, diabetic rats treated with a daily high dose (30 or 100 mg/kg) rather than 10 mg/kg oral cilostazol showed adequate free nerve endings compared to the control group and did not present mechanical allodynia. These findings suggest that dermal receptors that sense mechanical responses are also closely related to the activation of spinal dorsal horn microglia in response to persistent hyperglycemia. 

Diabetes-induced intraepidermal nerve fiber degenerative changes are a type of SFN. It may develop into a mixed neuropathy involving both small and large fibers [31]. However, small nerve fibers are mainly involved in initiating persistent hyperglycemic attacks. Multimodal mechanisms that induce SFN include inflammation, neurotoxic effects, and microvascular and tissue hypoxia. Persistent hyperglycemia or rapid plasma glycemic fluctuations produce intraneural inflammation, enhanced macrophage infiltration, increased cytokine levels in the sciatic nerve, and axonal degenerative changes of the intraepidermal nerve fiber [32]. It also causes diabetic microvascular complications through intracapillary malfunction and increased intercapillary distances to reduce nutrient transport and, subsequently, in a longstanding hypoxic endoneurial microenvironment [33,34]. Therefore, agents that decrease plasma glycemic levels exert anti-inflammatory effects and increased blood flow should be provided to avoid unexpected complications.

Cilostazol possesses the characteristics of antiplatelet, anti-mitogenic, and vasodilating properties, as well as anti-inflammation. The anti-inflammatory properties of cilostazol are not fully understood but may be closely related to reducing Toll-like receptor ligands-stimulated nuclear factor of kappa beta (NF-κB) transcriptional activity [35,36], pro-inflammatory factors [37] and inducible nitric oxide synthase [38]. Interestingly, our previous study also demonstrated that even daily oral low-dose (10 mg/kg) cilostazol for weeks was sufficient to produce significant changes to suppress microglial overactivation in the SDH of diabetic rats, but to restore the sodium channel dysregulation in the dorsal root ganglion necessitated an oral high dose of 100 mg/kg [20]. This indicates that peripheral nerves are less sensitive to cilostazol, thus presenting the need for a more effective dosage for meaningful small nerve-protective effects. However, the lack of significant trans-epidermal denervation in the diabetic dorsal trunk skin indicates the length dependence of SFN in the DM rat model, which is similar to one of the most common forms of pathological nerve degeneration in patients with diabetes. 

In DM rats, chronic hyperglycemia not only leads to peripheral nerve damage, as indicated by a significant reduction in P2X3- and CGRP-marked small sensory nerves in the DM rat paw epidermis, but is also a source of persistent and sufficiently intense stimulation, which could lead to central sensitization to certain stimuli. Typically, central or spinal sensitization denotes a form of functional synaptic plasticity that is use-dependent, leading to an enhanced responsiveness of nociceptive neurons in the central nervous system to normal or subthreshold afferent inputs [39,40]. The significant hypersensitivity towards mechanical stimuli, regardless of the apparent loss of peripheral small sensory nerve fibers, as indicated by P2X3 and CGRP expression in the hind paw epidermis, coupled with the increased expression of activated microglia in the SDH of DM rats, suggests that the observed mechanical allodynia in STZ-induced DM rats developed via central sensitization. The amelioration of mechanical allodynia in cilostazol-treated DM rats, in conjunction with significant reductions in the denervation of paw epidermal afferent nerves and microglial activation in the SDH, indicates that the medication could protect DM subjects from DPN and peripheral nerve disruptions via a central sensitization mechanism. Importantly, the significant reductions in detectable sensory nerve damage in the cilostazol-treated DM rats demonstrated the medication’s apparent ability to directly affect peripheral nerve integrity. Furthermore, the observable recoveries in the treated animals’ pain tolerance threshold against a mechanical stimulus indicated that cilostazol positively facilitated peripheral sensory nerve regeneration.

In contrast to mechanical allodynia, the thermal analgesia phenotype appeared to be positively correlated with the loss of CGRP- and TRPV-1-labeled peripheral afferent fibers, indicating SFN [39,41,42]. In the context of the present study, neuropathy is most likely a result of hyperglycemia-induced peripheral small fiber loss in the epidermis of DM rat paws, which further leads to desensitization in the periphery. Similar observations of late-phase hypoalgesia, particularly against target-heat stimuli, in the hind paws of STZ-induced DM rats have also been reported in conjunction with reduced expression of TRPV-1 in DRG neurons and the SDH [30,43].

## 5. Conclusions

In conclusion, the lack of a complete correlation between mechanical allodynia and denervation of peripheral afferent nerve fibers in the epidermis of DM rats indicated an association with central sensitization of the DPN phenotype. Although low-dose cilostazol appeared to be sufficient to ameliorate microglial hyperactivation in the SDH of STZ-induced DM rats in our previous study, the peripheral nerves required higher dosages of cilostazol to achieve the desired neuroprotective effects. Finally, oral cilostazol ameliorated hyperglycemia-induced peripheral small nerve fiber damage and peripheral epidermal denervation, thereby reducing the levels of diabetic neuropathic pain via a central sensitization mechanism.

## Figures and Tables

**Figure 1 medicina-59-00553-f001:**
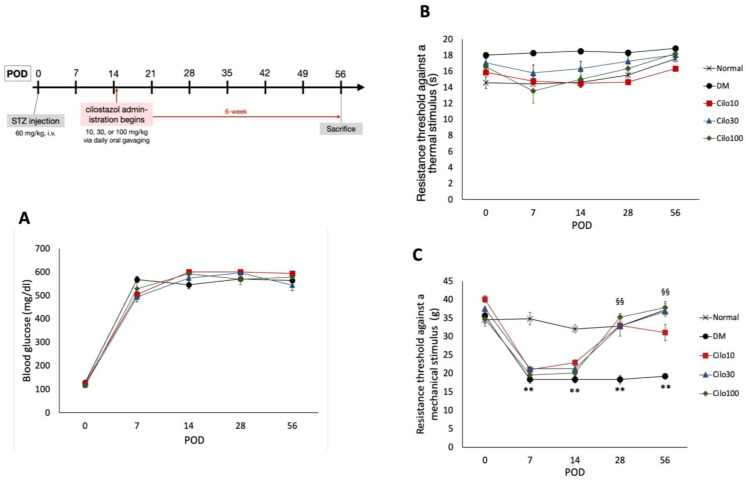
Experimental timetable and effects of cilostazol treatments on (**A**) blood glucose levels and hind paw withdrawal thresholds against (**B**) thermal and (**C**) mechanical stimuli. Scatter points represent SEM; error bars represent SE. ** Indicates significant differences at *p* < 0.001 compared to baseline values, and §§ at *p* < 0.001 compared to the corresponding DM value, standard single factor analysis of variance (ANOVA). POD, post-operation day; Cilo10, 10 mg/kg cilostazol; Cilo30, 30 mg/kg cilostazol; Cilo100, 100 mg/kg cilostazol.

**Figure 2 medicina-59-00553-f002:**
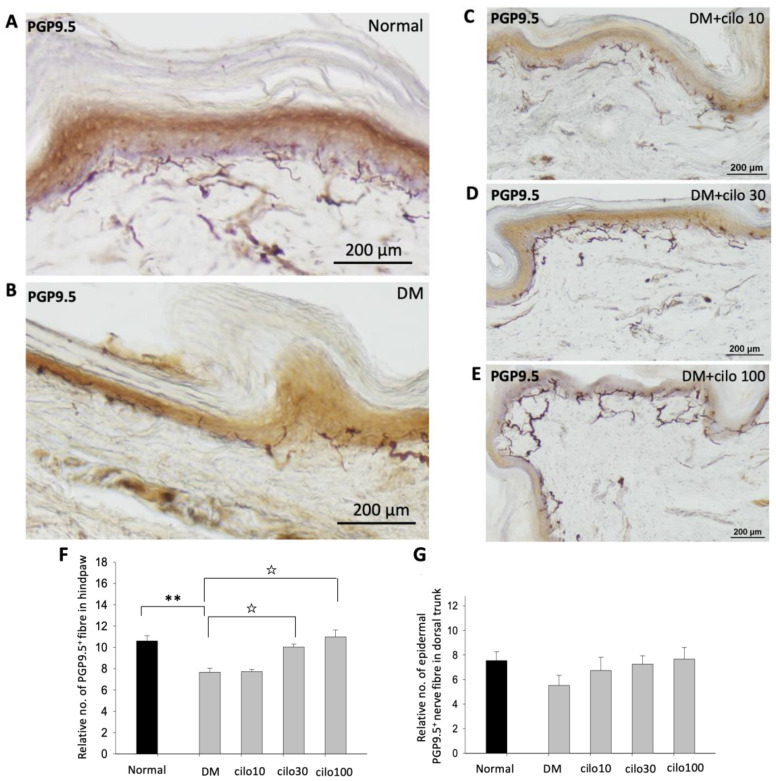
Ameliorated levels of PGP9.5-positive nerve endings in hind paws of cilostazol (cilo)-treated diabetic rats. PGP9.5 immuno-labeled sensory nerve fibers are indicated by white arrowheads in cryo-sectioned hind paw samples from (**A**) control rats, (**B**) DM rats, and DM rats receiving (**C**) 10 mg/kg (Cilo10), (**D**) 30 mg/kg (Cilo30), and (**E**) 100 mg/kg (Cilo100) daily dosage of cilostazol. Relative densities of intra-epidermal PGP9.5-positive nerve fiber (**F**) in the epidermal layer of hind paws or (**G**) in the epidermal layers of the back trunk skin are shown. Compared to the control group (**H**), reduction and swelling or shortening of PGP9.5-positive nerve fibers were observed in STZ-induced diabetic rats’ hind paws. (**I**) Intraepidermal PGP9.5-positive sensory nerve endings are indicated by white arrowheads, and black arrows indicate dermal nerve bundles in the foot pad. Epi, epidermis; der, dermis. Cilo10, 10 mg/kg cilostazol; Cilo30, 30 mg/kg cilostazol; Cilo100, 100 mg/kg cilostazol. Bars represent the average number of detectable labeled fibers in 5 mm skin sections. ** Indicates *p* < 0.01, and ☆ at *p* < 0.001, Mann–Whitney U test. Error bars represent SE.

**Figure 3 medicina-59-00553-f003:**
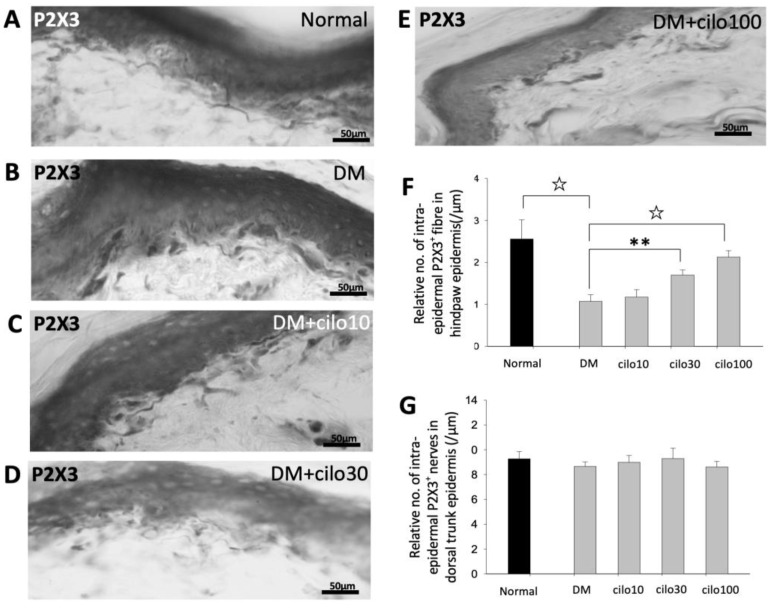
Increased levels of P2X3-positive sensory nerve endings in hind paws of diabetic rats treated with cilostazol (cilo). P2X3 immuno-labeled sensory nerve fibers are indicated by white arrowheads in cryo-sectioned hind paw samples from (**A**) control rats, (**B**) DM rats, and DM rats receiving (**C**) 10 mg/kg (Cilo10), (**D**) 30 mg/kg (Cilo30), and (**E**) 100 mg/kg (Cilo100) daily dosage of cilostazol. Relative density of intra-epidermal P2X3-positive nerve fiber (**F**) in the epidermal layer of hind paws or (**G**) in the epidermal layers of the back trunk skin is shown. Intraepidermal P2X3-positive sensory nerve endings are indicated by white arrowheads in the foot pad. Cilo10, 10 mg/kg cilostazol; Cilo30, 30 mg/kg cilostazol; Cilo100, 100 mg/kg cilostazol. Bars represent the average number of detectable labeled fibers in 5 mm skin sections. ** Indicates *p* < 0.01, and ☆ at *p* < 0.001, Mann-Whitney U test. Error bars represent SE.

**Figure 4 medicina-59-00553-f004:**
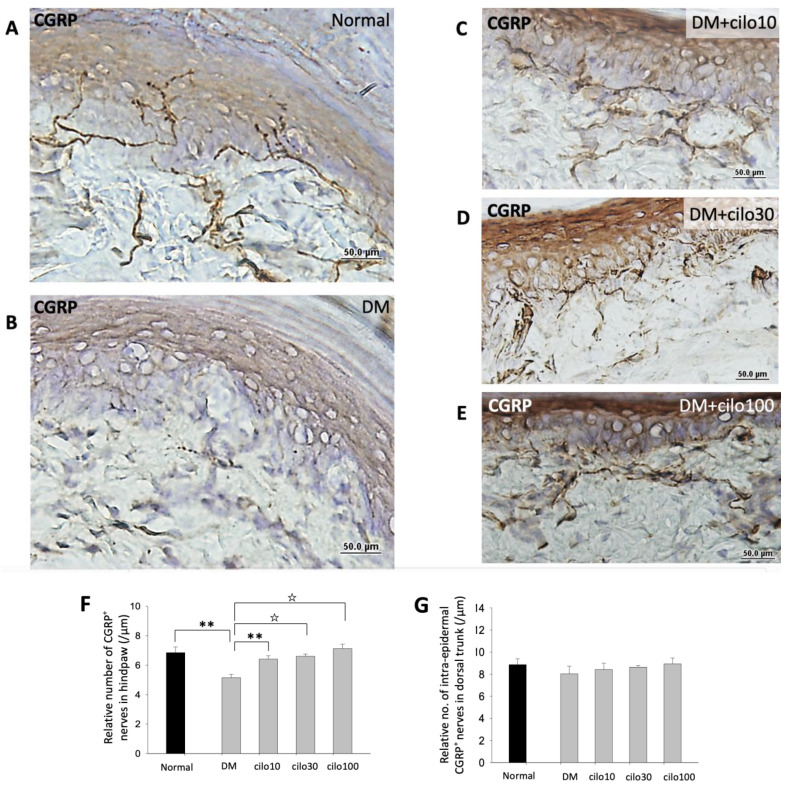
Increased levels of CGRP-positive sensory nerve endings in hind paws of diabetic (DM) rats treated with cilostazol (cilo). CGRP immuno-labeled sensory nerve fibers are indicated by white arrowheads in cryo-sectioned hind paw samples from (**A**) control rats, (**B**) DM rats, and DM rats receiving (**C**) 10 mg/kg (Cilo10), (**D**) 30 mg/kg (Cilo30), and (**E**) 100 mg/kg (Cilo100) daily dosage of cilostazol. Relative density of intraepidermal CGRP-positive nerve fibers (**F**) in the epidermal layer of hind paws or (**G**) in the epidermal layers of the back trunk skin is shown. Cilo10, 10 mg/kg cilostazol; Cilo30, 30 mg/kg cilostazol; Cilo100, 100 mg/kg cilostazol. Bars represent the average number of detectable labeled fibers in 5 mm skin sections. ** Indicates *p* < 0.01, and ☆ at *p* < 0.001, Mann–Whitney U test. Error bars represent SE.

**Figure 5 medicina-59-00553-f005:**
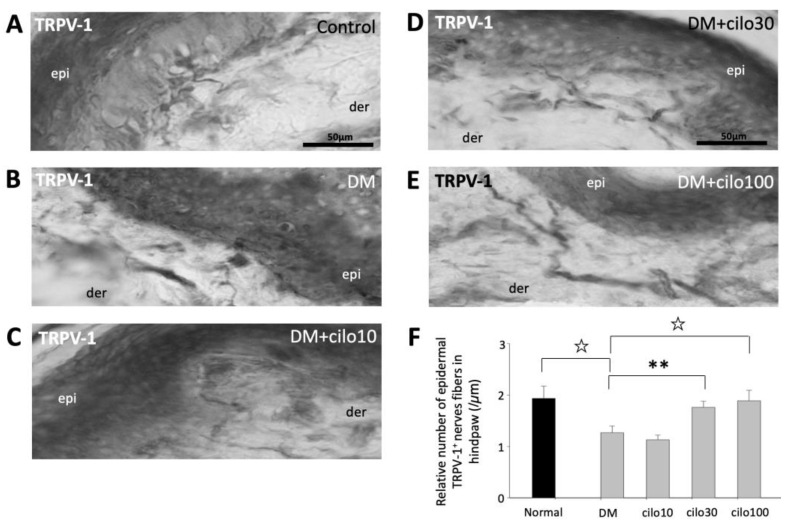
Ameliorated levels of TRPV-1–positive epidermal nerve endings in hind paws of diabetic rats treated with cilostazol (cilo). TRPV-1 immuno-labeled sensory nerve fibers are indicated by white arrowheads in cryo-sectioned hind paws samples from (**A**) control rats, (**B**) DM rats, and DM rats receiving (**C**) 10 mg/kg (Cilo10), (**D**) 30 mg/kg (Cilo30), and (**E**) 100 mg/kg (Cilo100) daily dosage of cilostazol. The relative density of intraepidermal TRPV-1-positive nerve fibers (**F**) in the epidermal layers of hind paws is shown. Cilo10, 10 mg/kg cilostazol; Cilo30, 30 mg/kg cilostazol; Cilo100, 100 mg/kg cilostazol. Bars represent the average number of detectable labelled fibers in 5 mm skin sections. ** Indicates *p* < 0.01, and ☆ at *p* < 0.001, Mann–Whitney U test. Error bars represent SE.

**Figure 6 medicina-59-00553-f006:**
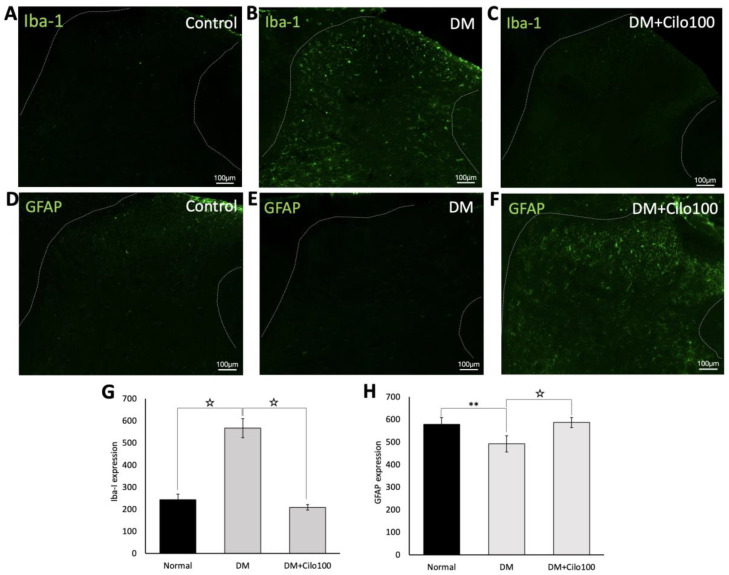
Reduced levels of microglia hyperactivation and increased levels of activated astrocytes in spinal dorsal horns of diabetic rats treated with cilostazol. Representative images of Iba-1 positive microglia (**A**–**C**) and GFAP positive astrocytes (**D**–**F**) expressions in spinal dorsal horns of (**A**,**D**) control, (**B**,**E**) DM, and (**C**,**F**) DM with cilostazol 100 mg/kg (Cilo100)-treated rats. Expressions of (**G**) Iba-1 positive microglia and (**H**) GFAP positive astrocytes in spinal dorsal horns are indicated as mean ± SEM for a minimum of five rats for each group. Expressions were measured in relative fluorescence levels. Error bars represent SE. ** Indicates significant difference at *p* < 0.01, and ☆ at *p* < 0.001, Mann–Whitney U test.

## Data Availability

The data presented in this study are available in Section 3. Result of this manuscript.

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
