# Peer review of "Peripheral Nerve Denervation in Streptozotocin-Induced Diabetic Rats Is Reduced by Cilostazol"

_medicina, 2023, doi:10.3390/medicina59030553_

Round 1

Reviewer 1 Report

In this study the authors investigated the mechanism of diabetic neuropathy (DPN) and cilostazol has been found to be effective to prevent peripheral nerve damage and ameliorate central sensitization as well.

The study investigated two aspects of neuropathy in the same time: the peripheral denervation and the central sensitization. Alteration of dorsal root ganglia and effect of cilostazol have been previously assessed in a recent publication.

 Comments to results

On Figure 2, the quality of figures are quite good, but improvement is necessary. Magnification of PGP9.5 -labeled sections from hidpad is low to evaluate them, higher magnification is recommended. In case of trunk samples, the image quality is generally low. Furthermore, here arrows show dermal fibers instead epidermal ones, which is appeared on part G, as bars. Parts F and G demostrate the summary results. Although results are understandable, but the exact meaning of bars are not clear here: bars represent the average number of fibers of which length? 1 mm, or 5 mm (I guess 1mm), please clarify on the figure, or in the legend.

The quality of images on figures 2, 4 and 5 are generally low. I understand that it is difficult to produce high quality images from such a thick sections, but this images add only few more information, I prefer not to show them, just the summary with bars, or somehow try to produce better images.

Comments to discussion

The different appearance of diabetic neuropathy in hindpad and trunk skin simply indicates the length-dependence of SFN in rat. It is similar to, but not “suggests”  a well-known, common form of human diabetic neuropathy, please modify the indicated sentence below accordingly.

„However, the lack of significant trans-epi-

320

dermal denervation in the diabetic dorsal trunk skin suggests that SFN in length-depend-

321

ent neuropathy is the most common form of pathological nerve degeneration in patients

322

with diabetes.”

Although the authors highlight that cilostazol improves neuropathy via central sensitization, a direct effect of this drug on peripheral nerves, such as rescuing from denervation is not emphasized.

Reviewer 2 Report

Dear Dr.,

Title: Peripheral nerve denervation in streptozotocin-induced diabetic rats is reduced by Cilostazol

Manuscript ID: medicina-2258333

Overall comments: Authors described in this manuscript: cilostazol effectively managing the symptoms of diabetic neuropathy in type-I model of rat. Further, the author claimed that it suppresses microglial hyperactivation and increased astrocyte expressions in spinal dorsal horn levels. The overall manuscript is written well and it has novelty in this field of research. However, some of the typographical errors and sentences need to revise in this manuscript.

Specific comments:

1.      The title of this manuscript word ‘Cilostazol’ must be ‘cilostazol’.

2.      The abstract is well written.

3.      Line no 94 to 99 author described ……..surgery to expose the right femoral vein and intravenous injection ……..daily. Why the author preferred the femoral vein for intravenous injection; normally tail vein injection is preferred.

4.      STZ 60 mg/kg; i.v. administration; what is the % mortality in this model and type-I diabetes incidence rate need to address in the methodology section.

5.      Section 2.3 order of explanation must be thermal pain response followed by mechanical stimuli associated with pain response with more clarity.

6.      On what basis author selected the dose of cilostazol 10, 30, and 100 mg/kg?

7.      There is some typographical errors were observed throughout the manuscript (example line 206: ……..10mg/kg…….).

8.      Figure 3-5; resolution needs to improve.

9.      Author Contributions: pattern must be rewritten as per the author's instruction in this journal.

10.  Numerous references are too old. Need to place recent references.

*****
